# The Mediating Role of the Social Identity on Agritourism Business

**Nesrine Khazami** [1,*] **and Zoltan Lakner** [2,*]

1   Doctoral School of Economics and Regional Sciences, Hungarian University of Agriculture and Life Sciences (MATE), 2100 Godollo, Hungary
2   Department of Food Chain Management, Institute of Agribusiness, Hungarian University of Agriculture and Life Sciences (MATE), 2100 Godollo, Hungary
*   Correspondence: nessrinekhazami@gmail.com or Khazami.Nesrine@phd.uni-mate.hu (N.K.); Lakner.Zoltan.Karoly@uni-mate.hu (Z.L.)

**Abstract:** There is a significant relationship between social capital, functional competences and social identity which forms the environment of rural tourism. This complexity was studied using the PLS-SEM approach, applying the initial corrected bias method based on direct questionnaire surveys among rural tourism entrepreneurs in Tunisia. The results of the bias-corrected primer model revealed that the entrepreneur's social identity mediated the link between social capital and functional competencies. Managerially, social capital supports rural lodge entrepreneurs in the process of defining their marketing strategy and optimizing the different components of their marketing mix, focusing on the differentiation of their products and services. A strong link within the entrepreneur's social capital network will encourage them to strengthen their social identity, leading to the enhancement of their different functional competencies.

**Keywords:** agritourism; entrepreneur; functional competencies; rural lodging; social capital; social identity





## 1. Introduction

The tourism sector is an important vector of economic progress all over the world, especially in developing countries [1]. The sector's potential can be further enhanced by collaboration with other branches of the national economy including agriculture [2]. Establishing closer links between tourism and agricultural production enhances the value-added content of the services offered by tourism, and it is an important potential opportunity for the diversification of economies, protection of cultural heritage, creation of new workplaces and utilisation of the workforce that would not be able to integrate into the social division of labour (e.g., rural women, elder generation, citizens with reduced capacity to work) [2]. All these favourable consequences of rural tourism development contribute to regional development in general and rural development in particular, especially in the case of less favoured regions [3]. As a summary, it can be stated that rural tourism is an important component of diversified, multifunctional regional development.

In the economy of Tunisia, rural and agritourism plays an increasing role. Entrepreneurs are aiming to diversify their product and service portfolios, offering an increasing set of services beyond the traditional aspects of tourism (e.g., guest houses, rural lodges, horse farms, horse riding tours). This strategy fits well within the general trends of global tourism development, because personalised, tailor-made offers of tourism by service providers are in line with the increasing demand for specific tourist attractions, orientating away from mass tourism toward specific services [4].

Of course, the increase in value-added content of tourism services offered by entrepreneurs demands considerable material investment, but first it requires a radical change in attitudes [5].

Paradoxically, the material development can often be achieved (e.g., renovation and reconstruction of buildings) faster than the changes in regulatory frameworks and attitudes of legislators and public service officials [4].

Rural tourism is tightly connected to the regional structures of interpersonal relationship among entrepreneurs. It is well documented, that these relationships shape.

Concerning the social capital of an agritourism entrepreneur, the social identity of an entrepreneur is presented as a reinforcement the development of their social capital. Individuals perceive themselves and are seen by others as part of a social group. The association of an individual within a specific social group also contributes to the formation of that social identity [6]. In the current study, it allowed us to see that the relationship between an owner of a rural tourist lodge and the development of a strong social identity permit them, quite often, to strengthen these functional competencies as well as contribute to making themselves known and creating an image of the enterprise. To the best of our knowledge, this is the first attempt to show these relationships in the case of agrotourism management.

In the current article, we sought more specifically to answer the following questions:

❖　What is the effect of entrepreneurial social capital on the development of entrepreneurial functional competencies?
❖　Is social capital an asset for better development of the functional competencies of an entrepreneur of rural lodgings?
❖　Does social capital facilitate the development of an entrepreneurial social identity?

To answer these questions, we considered it appropriate to first adopt a post-positivist approach centred on a literature review related to the development of a conceptual model and then a quantitative study to explain the phenomenon and analyse the data.

The objective of this article was to provide insight into the effects of social identity on the development of the relationship between social capital and the functional competencies of an entrepreneur. In this case, social capital was treated as an independent or explanatory variable; social identity and functional competencies are thus understood as dependent variables. The combination of the three concepts is still lacking in research [4,7]. This article is organised as follows. After the Introduction (Section 1), the theoretical framework and the hypotheses to be tested are presented in Section 2. Section 3 illuminates the methodological design for the empirical analysis. Section 4 presents the major findings and discussion. The final two sections include the conclusions and limitations.

## 2. Literature Review

### 2.1. Social Capital

It is well documented that social capital influences economic development via two paths: circulation and dissemination of information and creation of trust [8]. Both of these are essential to the development of rural tourism, because interpersonal relations contribute to the rapid proliferation of knowledge and know-how [9]. This process contributes to the strengthening of different enterprises and reduces transaction costs, because in this way, a considerable time and energy spent searching for information can be saved [8]. An entrepreneur with a wide range of connections based on his/her social capital has considerable additional competitive advantages, because they have rapid access to information for strategic planning of an enterprise as well as operative planning (e.g., costs of resources, formation of optimal decisions on procuration of services and living labour) [9]. On the other hand, the social capital contributes to the better positioning of the enterprise in mind for potential and actual consumers, contributing to the balance of supply and demand [10]. The economic role of social capital in the enhancement of trust among entrepreneurs is well proven in the literature [8].

### 2.2. Social Identity

The connection between entrepreneurs' identity and behaviour is a relatively lesser-known area [11]. In the opinion of Burke and Reitzes [12], there is a mutual reference context

between these factors: an entrepreneur with a special reference framework, linked to and determined by his or her identity, will utilise the same reference framework in the process of entrepreneurial decision making. From this follows that there is a continuous adjustment between the identity and the entrepreneurial interest [13]. Social entrepreneurial identity significantly influences the behaviour of the entrepreneur to create and exploit the business opportunity [14].

Based on the research in [11], three main types of entrepreneurial identities can be recognised: Darwinian, communitarian and missionary identities [11]. This typology is based on three dimensions: social motivation, source of self-assessment and edge of orientation. The Darwinian identity characterises the "typical (classic) entrepreneur", whose basic aim is the launching and maintaining of a strong and productive business [15]. For these entrepreneurs, the business is a way of self-fulfilment. The communitarian identity-oriented entrepreneurs have a strong commitment to the community, building and strengthening local communities and clusters of local entrepreneurs [16]. These entrepreneurs are relatively stable figures of local communities [17]. The missionary identity constantly searches for the place and role of the enterprise for the improvement of society. Therefore, their enthusiasm is thoroughly linked to social entrepreneurship [18]. In the opinion of Jones and al. [19], individuals with a social entrepreneurial identity need to distinguish themselves from another type, mainly profit-seeking entrepreneurs.

*2.3. Functional Competencies*

Previous work in the field of entrepreneurial skills has attempted to explain entrepreneurial success through the psychology of the entrepreneur mediated by his personality traits and motivation. This current of research, qualified as a psychological or "trait" approach, has given mixed results with traits and motivations, which they do not discriminate between successful and unsuccessful entrepreneurs [20]. One of the most stimulating consequences of this approach is the contribution to the emergence of a critical current, such as Gartner [21] suggested, with a more demanding interest in explaining the performance of the company created by the study of the behaviour of the entrepreneur. The emphasis is firmly placed on the entrepreneur's actions with their skills in following such as the ability to develop a business vision [22], the ability to identify business opportunities [23] and the ability to mobilise network resources. Gradually, the first typological work to classify the skills of entrepreneurs appeared. Herron and Robinson [24] formulated a typology of seven skills. The entrepreneur must be able to design products/services, evaluate the various functions of the company, understand its sector of activity and its trends, motivate its staff, create relationships with network business and plan and administrate the activities of the company to implant more opportunities. Based on the research of 134 owner–managers of SMEs, Chandler and Jansen [25] proposed a typology with three categories: entrepreneurial skills, managerial skills and technical–functional skills. The skills listed are ability to identify and exploit opportunities, ability to work intensely, ability to lead individuals, political ability to assert one's position in a business network and technical ability.

## 3. Research Model and Hypothesis Development

A considerable number of publications analyse the role of social identity [7,25] and social capital [4,10] in entrepreneurial decision making and behaviour. The role of social identity as a way to help people to know who they are, how they should interpret their links with others and how they must behave in given social circumstances has been analysed in detail [26]. Social identity is a rather simplifying concept, because in practice people possess numerous, simultaneously coexisting identities [27]. In addition, three different types of psychological recognition are appropriate and are due to the development of social capital: Darwinian identity, communitarian identity and missionary identity. Corresponding to Brewer and Gardner [28], the level of social identity suggests that the "distinction and individualism self-perception is the most characteristic feature of self -studies in western

psychology" (p.84). At the sub-group level, the community identities of individuals reveal major and essential combinations contained by the environment in which a person is in the right place. Finally, the missionary identities of individuals relate to the biggest important managerial combination. The previous studies recommends two crucial results concerning the relations amongst these three stages of psychological identity. Initially, the importance of a provided identity may differ in various circumstances [27]. Thus, whether persons interpret themselves mainly in conditions of a Darwinian, communitarian or missionary identity will be affected by a diversity of situational issues or circumstantial indications that bring out a provided amount of self-image. On the other hand, the psychological stimulation of these different self-representations produces distinct effects related to the perception that individuals have about themselves and others (their correspondences, their motivation to react in social circumstances and their assessing positioning concerning their activities).

Precisely when Darwinian identities are prominent, persons manage to interpret themselves and to interpret the options accessible to them in comparatively personal conditions. Consequently, they are expected to react in a relatively selfish manner. Therefore, when communitarian identities are relevant, people are expected to interpret themselves and the preferences accessible to them in conditions to react on the inner circle such as a component. In this condition, comparisons at the intergroup stage come to be relevant. Persons have a tendency to reflect regarding to the quality of their group related to other subgroups. Ultimately, when the missionary identity is relevant, persons are apt to interpret themselves as well as their comportment of choice according to its effect upon the commune. Notice that the results at the individual or group stage have a tendency to wane. The empirical assistance for these common theoretical proposals is significant. From an experimental study, the stimulation or creation of mutual identities of salient individuals improved their commitment to participate in mutual comportment. Various research from the previous two decades, for example, have demonstrated that merely boosting the salience of a combined social identity is sometimes necessary to strengthen a person's commitment to participate in its establishment or preservation. Additional research has demonstrated that mutual recognition showed that a personal distinction variable is similarly expected as comportment [28,29]. In summary, the findings of various experimental research demonstrate that missionary recognition enables the commitment to provide the results at the mutual stage in a diversity of situations, varying from small investigational groupings to large businesses and social groupings. Taken together, the consequences of this research join the assumption that circumstantial social indications cause the missionary identities of salient individuals to increase their tendency to become involved in collectivist manners—the types of manners, in return, maintain the social capital. Certainty, the authenticity of this general conclusion is reinforced by these findings that have been noted in numerous studies, including a large variety of chosen predicament circumstances and using a range of various investigational strategies and methods of missionary identity. Based on the discussions above, our hypothesis is as supports:

**Hypothesis 1 (H1).** *Social capital positively affects the social identity of an entrepreneur.*

Identifying an individual with a given social group (a team, an age group, a professional identity, etc.) means that he/she (in the majority of cases) considers him/herself as belongs to it [29].

To study this process, current research draws on two fundamental theoretical currents. One of the postulates of the social identity theory [11] is the presupposition that an individual tends to identify him/herself with a given social group, if this group is considered for him/her as an attractive one. Social categorisation theory [30] specifies how individuals distinguish between different social categories and classify themselves as members of a group. Recent studies have revealed a "hidden face" of this identification [31] and have proposed to renew the research by considering other forms of identity construction [32]. The complexity of identity can be well characterised on the basis of the categorisation of

this identity [33]. Indeed, it appears that individuals do not necessarily identify with a specific social group. Taking the case of a team, there are four scenarios: a strong identification (the team is a reference group), a schizo-identification (the individual simultaneously refers to other social groups), disidentification (his level of identification is low) and sous identification (he is defined in opposition to this social group). This conceptual framework makes it possible to understand the lack of specific management and the articulation between individual and collective skills. The diversity of skills is potentially a source of tension through discrimination between social groups (for example, between professions in a project team), sometimes to the point of creating rejection by the team (disidentification). Some surveys showed that communication becomes more difficult, to lead conflicts and rotations [34]. If individuals are not able to mutually recognise their contribution of skills and to confront their points of view, there is little chance that they will mobilise a collective competence. In addition, there is a risk of schizo-identification, since these people are called upon to constantly refer to other attractive social groups (for example, their professional community). As confirmed by a longitudinal survey of multiple operational teams [35], the manager is then called upon to make a permanent effort to integrate and unite the team, under penalty of a dilution of his identity and, consequently, of an underutilisation of its collective skill potential. This is not to deny the differences but to show that they are legitimate and useful [36]. He must also overcome the difficulties of understanding between team members, since different experiences of professional socialisation and social categorisation create specific languages and vocabularies [37].

Hence, the following hypotheses:

**Hypothesis 2 (H2).** *Social capital positively affects the functional competencies of an entrepreneur.*

**Hypothesis 3 (H3).** *Social identity positively affects the functional competencies of an entrepreneur.*

**Hypothesis 4 (H4).** *The relationship amongst social capital and functional competencies of an entrepreneur are mediated by the social identity.*

In brief, the components and links including in the model suggested in this article are introduced in Figure 1.

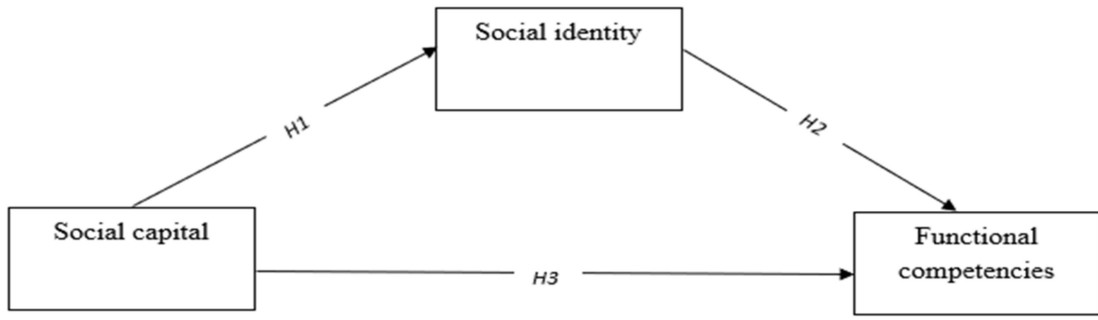

H4: Social capital -> Social identity -> functional competencies

**Figure 1.** Conceptual model (Source: own compilation, 2021).

## 4. Methodology

### 4.1. Research Instrument

This study examined the relationship between social capital, social identity and functional competencies in the context of agritourism in Tunisia. The questionnaire for social capital was measured using ten items from Nahapiet and Ghoshal [38], while the

components of an entrepreneur's social identity were adopted from Sieger and al. [39] and were measured by ten items. Lastly, functional competencies were measured using nine items adopted from Lichtenstein and Lyons [40]. The relevant elements are included in Appendix A.

### 4.2. Sample and Data Collection

An electronic survey was adopted to verify the conceptual model identified in Figure 1. The population was defined as entrepreneurs from Tunisian rural fora who had spent at least two years in business in Tunisia.

Due to the coronavirus pandemic, we contracted with potential respondents via social media. The online data collection was realised via Google Drive. Only completely filled out questionnaires were accepted by the system, which is why there were no vacancies in the input data. The number of respondents was relatively low, our sample can be considered as representative and acceptable (nearly 60% coverage rate) because business in rural tourism is rather new in Tunisia.

The demographic characteristics of the participants are presented in Table 1. The random sample error was approximately 79.3% (95% confidence interval). Likert scales were applied, where 1 meant the lowest level of acceptance (total refusal) and 5 indicated the highest level of acceptance.

**Table 1.** Descriptive analysis (*N* = 100).

| Demographic Characteristics | Category | Percent |
|---|---|---|
| Gender | Male | 63 |
| | Female | 37 |
| Age | 18–34 | 14 |
| | 35–49 | 55 |
| | 50–60 | 31 |
| Education | High school | 42 |
| | Bachelor's degree | 53 |
| | Postgraduate (+5 years) | 5 |
| Income/monthly/person (dt *) | 1200–4500 dt | 77 |
| | 4600 dt< | 23 |

* 1DT (Tunisian Dinar) = 0.31 EURO.

### 4.3. Analytical Methods

As demonstrated in Figure 1, that there was a significant relationship between social capital and functional competences. This relationship was moderated by social identity, which is why there was a significant relationship between social capital and social identity as well as between identity and functional competences. This system of relationships can be quantified by structural equation modelling.

Partial least squares structural equational modelling (PLS-SEM) was used to test the system of hypotheses outlined above. The PLS-SEM approach has numerous advantages over other algorithms such as sample size constraint, high efficiency, precision and assessment accuracy and flexible modelling environment. It is important to emphasize, that this approach does not demand the normality of input data [41]. In addition, SEM is a mixture between two potent statistical methods: exploratory factor analysis and structural path analysis,s which allows simultaneous evaluation of the measurement model and the structural model [41].

## 5. Results and Discussion

### 5.1. Demographics of Participants

Overall, 107 self-administered questionnaires were delivered to rural accommodation entrepreneurs in the various rural regions of Tunisia. One hundred questionnaires were completed. The findings of the descriptive statistics are presented in Table 1. It is well known that Tunisia, similar to other Muslim countries, is a male-dominated society where the majority of managerial positions are occupied by men. This fact was well reflected in the gender structure of respondents: nearly two-thirds of respondents were men. Our selection criteria decreased the chances of younger persons participating in the sample; this is why the majority of respondents were middle-aged persons. In this way, we were able to collect pieces of information from relatively experienced entrepreneurs. This age characteristics of the sample were in line with the international trends: the typical age of start-up entrepreneurs is between 25 and 40 years of age in the US and Canada at the time of the start-up [38].

Knowing the descriptive analysis, it can be seen that all entrepreneurs were graduates of higher education. Indeed, 53% of entrepreneurs had university degrees. Forty-three percent of entrepreneurs completed high school, while 5% of entrepreneurs had postgraduate degrees. This is explained by the encouragement and investment incentives granted by the public authorities to this category of entrepreneurs holding university degrees in order to reduce the problem of unemployment among executives, on the one hand, and to create new employment through investments and businesses created on the other hand. Most respondents earn between 1200 and 4500 TND (77%). This confirms that the investment generally creates jobs and contributes to increasing the company's income.

The successful running of a rural tourism business demands a high level of diversified competencies, which is why the share of accomplished high school or higher qualifications has been well above the Tunisian average: 53% of respondents had at least a BSc/BA degree. This high share of intelligentsia shows another interesting phenomenon: as our interviews with entrepreneurs and public service professionals highlight, rural tourism-based entrepreneurship is an important strategy for qualified people who could not find any other source of income. The income generated by rural tourism, in most cases, has not been higher than 4500 Tunisian Dinar (approximately 1400 Euro). Taking into consideration the considerable costs of tourism enterprise and the high risks associated with this type of business (e.g., market fluctuations), this can be a modest income.

### 5.2. Factor Analysis

A first factor analysis was conducted. For the four elements filled out, if one of the factors were missing, they were therefore removed (i.e., Darwinian identity (SID3), missionary identity (SIM4), functional competencies (FC5) and functional competencies (FC 7)). With these results, the model presented in Figure 1 could be tested via a structural equation model. Nevertheless, the partial least squares (PLS) technique applied requires information not only on the structural model but also on the measurement model. Indicators with loads greater than 0.7 are generally appropriate, but those greater than 0.6 may possibly be maintained in newly established procedures. Subsequent to these criteria, the elements FC3 and FC6 were eliminated from the construct of functional competencies.

### 5.3. Assessment of Model Using Partial Least Squares Structural Equation Modelling

The reliability statistics for every construct are exhibited in Table 2. The results of Cronbach's alpha indicate that the alpha value for all variables was higher than 0.70; this indicator shows a high level of internal reliability [42].

Table 3 summarises the most important characteristic features of the distribution of variables. Obviously, based on these statistics, the different constructs show normal distribution.

**Table 2.** Reliability and validity of the constructs.

| Constructs | Dimensions | Indicators | Loadings | Cronbach's Alpha | Rho_A | Composite Reliability |
|---|---|---|---|---|---|---|
| Social capital | | | | 0.955 | 0.954 | 0.957 |
| | Structural capital | SCS1 | 0.877 | 0.848 | 0.864 | 0.922 |
| | | SCS2 | 0.812 | | | |
| | | SCS3 | 0.734 | | | |
| | Relational capital | SCR1 | 0.597 | 0.858 | 0.884 | 0.905 |
| | | SCR2 | 0.859 | | | |
| | | SCR3 | 0.892 | | | |
| | | SCR4 | 0.821 | | | |
| | Cognitive capital | SCC1 | 0.809 | 0.914 | 0.911 | 0.937 |
| | | SCC2 | 0.935 | | | |
| | | SCC3 | 0.844 | | | |
| Social identity | | | | 0.954 | 0.984 | 0.960 |
| | Darwinian identity | SID1 | 0.935 | 0.889 | 0.915 | 0.933 |
| | | SID2 | 0.815 | | | |
| | | SID4 | 0.924 | | | |
| | Communitarian identity | SIC1 | 0.685 | 0.727 | 0.754 | 0.846 |
| | | SIC2 | 0.751 | | | |
| | | SIC3 | 0.871 | | | |
| | Missionary identity | SIM1 | 0.875 | 0.949 | 0.951 | 0.967 |
| | | SIM3 | 0.847 | | | |
| | | SIM4 | 0.939 | | | |
| Functional competencies of entrepreneur | | FC1 | 0.817 | 0.789 | 0.855 | 0.868 |
| | | FC2 | 0.786 | | | |
| | | FC4 | 0.819 | | | |
| | | FC8 | 0.655 | | | |
| | | FC9 | 0.677 | | | |

**Table 3.** Skewness and kurtosis.

| Variables | Mean | SD | Excess Kurtosis | Skewness |
|---|---|---|---|---|
| Social capital | 4.625 | 1.102 | 0.897 | −1.582 |
| Social identity | 3.922 | 1.088 | 0.720 | −1.433 |
| Functional competencies | 4.230 | 0.879 | 1.485 | −1.470 |

The analysis shows that the correlations amongst every construct verified the discriminant validity test [43]. Hatcher [44] suggests to verify it during the interval's confidence of the correlation.

The average variance extracted (AVE) expresses the portion of the variance, explained by the construct, which is why it can be applied as an indicator of convergence validity. An AVE exceeding 0.5 is generally adequate. This criterion was satisfied by all the constructs applied in the model. This fact supports the applicability of the construction (Table 4).

The value of $R^2$ indicates that social capital explained 56% of the variance of the dependent variable. The findings of the PLS-SEM path showed a significant positive relationship ($\beta = 0.462$ and *p*-value = 0.00) between social capital and social identity. This fact proves the H2 hypothesis. These results are in line with findings of other authors (e.g., [27,45,46]). Similarly, a positive and significant relationship ($\beta = 0.161$ and *p*-value = 0.083) could be proven between social capital and functional competencies, so H1 is accepted. These results agree with the findings of Tansley, C. and Newell, S. [47]; Henley, A. [48]; McCallum, S. and O'Connell, D. [49]; Khazami, N. and Lakner, Z. [4]. This fact supports the idea that

the functional skills of an entrepreneur are fostered by the development of different social identities and activities when starting a rural business. Analysis of the results shows that the structural linkage between social identity and functional skills is positive and significant (β = 0.604 and *p*-value = 0.000). Thus, hypothesis H3 is accepted.

**Table 4.** Discriminant validity test.

| | Functional Competencies | Social Capital | Social Identity |
|---|---|---|---|
| Functional competencies | **0.741** | - | - |
| Social capital | 0.552 | **0.833** | - |
| Social identity | 0.605 | 0.540 | **0.872** |

The AVE is presented in bold (*p* < 0.05).

Our model highlights the importance of the formation of social identity and social capital (Table 5). From this follows that the state should promote such governmental, municipal or civic organisations which could contribute to the enhancement and strengthening of social identity. Our results are in line with findings of another actors, e.g., Brändle, L. and et al. [50]; Chan and et al. [51]; Liñán [52]. The explanatory power of social capital rose to 73% when we tried to determine the mediating role of social identity. The findings of the PLS-SEM showed that social identity mediates the relationship between social capital and functional competencies (β = 0.268 and *p*-value = 0.000); therefore, H4 is accepted. This finding highlights that social capital contributes in a measurable way to the enhancement of the functional competencies of entrepreneurs.

**Table 5.** Summary of the assessment of the direct and indirect effect of social identity.

| Total Effect (SC-> FC) | | Direct Effect (SC-> FC) | | | Indirect Effects of (SC-> FC) | | | | |
|---|---|---|---|---|---|---|---|---|---|
| Coefficient | *p*-Value | Coefficient | *p*-Value | | Coefficient | SD | t-Value | *p*-Value | BI (5%, 95%) |
| 0.435 | 0.000 | 0.161 | 0.083 | H: SC->SI->FC | 0.268 | 0.052 | 5.135 | 0.000 | 0.190, 0.365 |

At this level, we tested the type of mediation according to Baron and Kenny [53]. The direct and indirect effects were substantial (Table 6). This fact highlights that social capital boosts the effect of social identity in formation and exploitation of functional competences. Put in plain English: a high level of "social capital" enhances the positive effect of the social identity of an entrepreneur's functional competences.

**Table 6.** Summary of the results of the type of mediation (social identity).

| Direct Effect Sign | Indirect Effect Sign | Product Sign (Direct Effect X Indirect Effect) | Type of Mediation |
|---|---|---|---|
| +0.161 | +0.268 | +0.038 | Complementary |

This research provides an answer to the doubts of the owners of rural lodges on the important role of social capital in the development of their businesses. Indeed, the different relationships allow entrepreneurs to freely express their knowledge, which makes it possible to influence the economic activity of the entrepreneur but also more broadly on local tourism development. When a trust relationship is established between all the parties concerned, it influences their strategic behaviour, the setting-up of new investments or even the agreement on the price level exercised.

## 6. Conclusions

It is well known that the social capital of an entrepreneur can promote the economic performance of his/her activity, because his/her interpersonal relationships facilitate the flow of information and promote access to new material (raw materials, capital) or immaterial (knowledge) resources. Social identity plays an important role in the acknowledgement of an individual's place and role in the social structure of local society. The connection of an individual with a specific social group contributes to the formation of social identity, too.

The novelty of our study was highlighting the mutual role of social capital and social identity as factors enhancing social competition.

The findings of our study are well supported by our personal experiences, obtained in the process of qualitative analysis of strategy and the actual behaviour of rural tourism entrepreneurs, because a stronger association via social capital would encourage the development of functional competencies that already exist in the entrepreneur. These strong links are beneficial in terms of transfer of tacit knowledge, exchange of specific information and know-how.

Traditional enterprise-related literature has mainly focussed on strong connections. At the same time, weak ties are important, too [54], which are available to them to improve social cohesion and access different pieces of information, in this way improving functional competencies.

The current research has contributed to the enrichment of the theoretical knowledge on behaviour of an entrepreneur in agritourism by proposing and testing a model which was based on social capital as well as on the identity approach shaping the competences of the entrepreneur. The understanding of the mutual relationships between identity, social capital and functional competences can help to enhance the efficiency of policies focussing on upgrading the tourism-related entrepreneurial sector.

## 7. Limitations

The results of the current study supply a relatively reliable picture of the factors shaping the functional competences of rural tourism related entrepreneurs in Tunisia. At the same time, we must see that there are considerable limitations in the current work. The most important of these are as follows: (1) Relatively low number of respondents. Increasing the number of respondents could be an important step towards the enhancement of the reliability of the results. (2) The functional competences were influenced by numerous factors that were not analysed in the current study. A more detailed analysis of factors influencing the functional competences and analysis of the role of functional competences in the success of rural tourism business should be topics of future research work.

**Author Contributions:** Conceptualisation, N.K.; methodology, N.K.; validation, Z.L.; investigation, N.K.; data analysis, N.K.; writing—original draft preparation, N.K.; writing—review and editing, Z.L.; visualisation, Z.L.; supervision, Z.L. All authors have read and agreed to the published version of the manuscript.

**Funding:** This research received no external funding.

**Institutional Review Board Statement:** Not applicable.

**Informed Consent Statement:** Not applicable.

**Data Availability Statement:** Excluded.

**Acknowledgments:** This research was supported by the Hungarian University of Agriculture and Life Sciences (MATE) and a Stipendium Hungaricum Scholarship.

**Conflicts of Interest:** The authors declare no conflict of interest.

## Appendix A.

**Table A1.** List of items used to measure the variables.

| Variables | Items | |
|---|---|---|
| Social capital | For you, the number of social ties with business experience in agritourism is very important. | Nahapiet & Ghoshal (1998) |
| | For you, work experience in the tourism sector is very important. | |
| | For you, being village leaders is very important. | |
| | For you, willingness to exchange employment and investment information is very important. | |
| | For you, willingness to exchange money and other assets is very important. | |
| | For you, confidence in family and friends for strong support in a crisis is very important. | |
| | For you, trustworthiness to family and friends is very important. | |
| | For you, encouraging young people to become independent by operating a business is very important. | |
| | For you, paying close attention to and admiring successful entrepreneurs is very important. | |
| | For you, attitude towards employment in tourism/hospitality is very important. | |
| Social identity | For you, the opportunity to create economic value and to create personal wealth over time has been an important driving force. | Fauchart & Gruber (2011) and Sieger et al., (2016) |
| | For you, the focus on profitability is very important. | |
| | For you, success is that your business shows better financial performance compared to competitors. | |
| | For you, your main motivation is to show your personality traits as an entrepreneurship. | |
| | For you, your main motivation is related to offering a good and novel product that you know people have use for. | |
| | For you, to be true to the original idea and deliver products of high quality to your customer segments, is most important. | |
| | For you, success is that your products work well for those that are supposed to use them. | |
| | For you, the main motivation is that through your firm, you can pursue values that are important to you or a particular cause. | |
| | For you, success is that the firm can contribute to changes that make society a better place. | |
| | For you, it is important to you that you manage to show that there are other and better ways to do things in accordance with your values. | |
| | For you, eco- entrepreneur responsibility has been an important driving force to create your agritourism business. | |
| Functional competencies | For you, a business in agritourism helps the ability to take responsibility for solving a problem. | Lichtenstein and Lyons (2001) |
| | For you, a business in agritourism helps the emotional ability to cope with a problem. | |
| | For you, a business in agritourism helps the ability to think critically. | |
| | For you, a business in agritourism helps the ability to cooperate with others, networking and utilising contacts. | |
| | For you, a business in agritourism helps the ability to reflect and to be introspective. | |

**Table 1.** *Cont.*

| Variables | Items |
|---|---|
| | For you, a business in agritourism helps the ability to recognise market gap, exploit market opportunity. |
| | For you, a business in agritourism helps the ability to operate a business and strategic planning. |
| | For you, a business in agritourism helps the ability to set personal goals, reach them and set new ones. |
| | For you, a business in agritourism helps the ability to make a persuasive communication and negotiation skills. |

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
