# Peer review of "The Mediating Role of the Social Identity on Agritourism Business"

_sustainability, doi:10.3390/su132011540_

Round 1

Reviewer 1 Report

The paper is interesting. Yet, it needs to be extensively amended:

  • The text presents repetitions, typing mistakes, sentences interrupted that make no sense as they are. I wonder if the Authors have revised it before submission.
  • The English form needs improvement: its poorness is evident even to a non-mothetougue as I am.
  • the first 8-9 pages explain extensively the theories about the figure of the Entrepreneur. This part could be reduced, at east in lenght, without loosing its meaning. This bibliographic excursus is very interesting, but too long.
  • the part devoted to the explanation of results is, conversely, very short, and a more thorough discussion is needed. 

Author Response

Dear Madam/ Sir,

We have taken into consideration all the remarks of the reviewer, and we have made the necessary changes.

We changed the part of the literature review, and we reduced the content of it and the references. We improved the English language and the style of the article. Also, we explained more about the part of the result.

Thank you.

Reviewer 2 Report

In the manuscript the authors presented an important point, although the manuscript has some drawbacks.

Main remarks:

1. Title - is correct, the title corresponds to the content of the presented manuscript.

2. Intruduction/Literature review - what is the research gap? Introduction / literature review should be extended with the factors of the development of agritourism in the present times. What was the definition of agritourism adopted by the authors for the purposes of the presented manuscript? What are the directions of agritourism development nowadays? Are innovations needed to expand the offer of agritourism farms? What are the authors of this opinion? Suggested publications: doi.org/10.3390/agriculture11050458, http://repo.lib.sab.ac.lk:8080/xmlui/handle/123456789/908, doi.org/10.1016/j.landusepol.2017.03.002, doi.org/10.3390/su12124858

3. Methodology - incorrect numbering of chapters (4.2, 4.4. ...).

4. Results - why exactly 100 people were surveyed? Moreover, a very small sample of the people surveyed. After table 5 there is table 35. In my opinion, interesting taxonomic measures can be developed for such research results, while adopting specific variables, e.g. social ones.

5. Conclusions - in your conclusions, please also answer the following questions:
• what are the directions for the future?
• what are the research gaps?
• what is new to this manuscript?

The presented manuscript is very interesting. This applies to the current situation in agritourism.

Author Response

Dear Madam/ Sir,

We have taken into consideration all the remarks of the reviewer, and we have made the necessary changes.

We explained more the part of the introduction as you mentioned using your recommendations, and we made some modifications. We modified the numbering of title 4. About the part of results, we explained why we used only 100 entrepreneurs in this study. We modified the number of table 6. We developed more the part of results. For the part of the conclusions, we answered your questions.

Thank you.

Round 2

Reviewer 1 Report

The paper has been improved in comparison to the first version. Yet,  several poits are still open:

  • the English form still leaves something to be desired. I strongly suggest the Authors to submit the text to a mothertongue translator.
  • the paper is still unbalanced: seven pages are devoted to a very interesting theoretical excursus, but very little is said about the content of the questionnaires, how they were administered, the areas in which the research took place. The statistical analysis is on the contrary well explained.
  • Given the above mentioned flaws, the resuls and the conclusions are only in part clear.

Author Response

Dear Madam/ Sir,

We have taken into consideration all the remarks of the reviewer, and we have made the necessary changes.

We corrected the English language and we tried to make a balance between the parts.

Thank you.

Reviewer 2 Report

Accept in present form! Good luck 

Author Response

Thank you for your acceptance. This is our last version of the article.

Thank you.

Round 3

Reviewer 1 Report

I am very sorry but I have to insist: this paper must be seen by an English mothertongue, with at least some experience on this scientific topic. As it is, it is not acceptable at all. Many sentences were evidently written by a person with a rough knowledge of English, and I report a few examples for the two first pages:

In Abstract, line 7: 

the sentence "entrepreneurs in process of definition their marketing strategy and optimization the different components of marketing mix” lacks articles and conjunctions. Maybe it could be changed to “in defining their marketing strategy and optimizinng the different components of the marketing mix”.

In Abstract, line 9: in the sentence "will encourage her to strengthen her social identity" the word "Her” makes no sense. It is singular femal. This means that all the work refers only to one single female entrepreneur!

Page 2, line 4: the sentence"can be often achieved...than"  makes little sense as it is: maybe it should be "can be more often achieved...than".

Page 2: in the paragraph beginning with "The rural tourism..." the second sentence is interrupted: "It is well documented, that these relationships shape the...". WHAT DO THEY SHAPE? Did the authors read the paper before submitting it?

Page 2, line 16: “to image creation of the enterprise” makes little sense, if "image" is the verb. Maybe the meaning is : “to create the image of the enterprise”.

Page 2, line 36: “after all” can be cancelled.

Some other flaws deal with references (Reference 28 is reported as Brewer and Gardner at page 4, line 7, while it is just Brewer in bibliography) , as well as the literature review and methodological design:

In Paragraph 3 “Research Model and Hypothesis Development”, in the sentence “disidentification (his level of identification is low) and sous identification (he is defined in opposition to this social group)”, It appears  that the Authors have inverted the two concept. In fact, “Disidentification occurs when an individual defines him or herself as not having the same attributes or principles that he or she believes  define  the  organization”, while  "neutral identification is based on the explicit absence of both identification and disidentification with an organization" . This, at least, according to the Authors quoted in Reference 33.

The same Authors published two papers in Sustainability a few months ago, on a similar topic: "Influence of Social Capital, Social Motivation and Functional Competencies of Entrepreneurs on Agritourism Business:
Rural Lodges" and "Influence of Experiential Consumption and Social Environment of local tourists on the Intention to revisit Tunisian Guesthouses: Mediating role of involvement in the experience", and these are good papers.

So, why this difference? what happened?